# High-throughput assessment of the behavioral responses to toxic organic solvents in *Caenorhabditis elegans*

**Masahiro Tomioka**[ID]*

Division of Industrial Toxicology, Research Center for Chemical Information and Management, National Institute of Occupational Safety and Health, Kawasaki, Japan

* tomioka-masahiro@h.jniosh.johas.go.jp

## Abstract

Novel chemical compounds are continuously being developed for use in various industries and daily life. Workers in these industries assess and avoid chemical hazards based on published information about chemical toxicities. However, avoiding the hazards associated with chemicals with unknown toxicity is difficult. Therefore, understanding the toxicities of chemicals in a high-throughput, multifaceted manner is essential. In this study, I developed a high-throughput method for assessing chemical toxicities through quantitative measurement of behavior in *Caenorhabditis elegans*. I determined the acute response to 30 organic solvents, including alcohols, cellosolves, ethers, ketones, and acetate esters, which are widely used in industries, with motility as an endpoint. Exposure to 0.5%–6% organic solvents caused a dramatic decrease in locomotion speed. The adverse effects of organic solvents on motility were proportional to the lipid solubility of the chemicals, similar to the positive relationship between the anesthetic effects of volatile organic chemicals and their lipid solubility in organisms, including humans. In addition to their effects on motility, organic solvents affect posture during locomotion in different ways depending on the chemical's functional group. Solvents with hydroxyl groups, such as alcohols and cellosolves (0.5%–3%), reduced the amplitude of body bending, whereas solvents with ketone groups, such as ketones and acetate esters (0.5%–4%), increased it during undulatory locomotion. In addition, organic solvents caused changes in chemotaxis plasticity based on the association between starvation and chemical signals at concentrations lower than those that affect locomotion. This study describes a high-throughput method for acute chemical toxicity testing and provides new insights into behavioral responses to organic solvents that are toxic to humans and other animals.

## Introduction

Organic solvents are used in various industries as well as in our daily lives. Many organic solvents are amphiphilic and can penetrate the epithelium, thus posing a risk of health hazards to organisms, especially workers in industries. Commonly used organic solvents, such as alcohols, ketones, and ethers, can cross the blood–brain barrier and cause neurotoxicity when

**Data availability statement:** All relevant data are within the manuscript and its Supporting Information files.

**Funding:** The author(s) received no specific funding for this work.

**Competing interests:** The authors have declared that no competing interests exist.

exposed to high levels of these chemicals. For example, low concentrations of benzyl alcohol are generally used in daily life, such as in foods, drugs, and cosmetics. However, benzyl alcohol intoxication has been reported in workers using paint strippers containing high concentrations of benzyl alcohol [1,2]. Intoxicated patients collapsed during paint stripping in a state of unconsciousness, and severe damage to the central nervous system has been reported [1]. To reduce hazardous chemical risks, a hazard assessment has been performed using information on chemical toxicities, such as the Globally Harmonized System of Classification and Labelling of Chemicals (GHS) [3,4], which classifies chemicals based on accident cases and toxicity studies on animals. However, many chemicals have unknown hazards. Benzyl alcohol had not been classified as a chemical with specific target organ toxicity to the brain before serious accidents occurred in Japan. To predict chemical hazards and prevent accidents before they happen, it is important to fundamentally understand the toxic effects of chemicals and the mechanisms of their adverse effects on organisms.

External chemicals interact with proteins and lipids on plasma membranes, as well as extracellular biological molecules, such as the extracellular matrix, and can cause toxic effects. Furthermore, chemical toxicities occur through the penetration of chemicals into cells, where they subsequently interact with biological molecules. Cell permeability depends on the chemical's physicochemical characteristics, including molecular size, melting point, and lipid solubility. The externally incorporated chemicals interact with biological molecules, such as proteins, lipids, and DNA, to exert their toxic effects. Exposure to high levels of organic solvents causes acute toxicity to the brain, followed by disturbances in consciousness, similar to the effects of volatile anesthetics. More than 120 years ago, Meyer and Overton independently reported that the anesthetic effects of volatile organic chemicals increase with increasing lipid solubility of the chemicals [5,6]. Subsequent studies have shown that increased lipid solubility elevates passive transport into cells through the plasma membrane, which is constituted of a lipid bilayer, and could affect the function of multiple lipids and proteins within the cell membrane, such as ion channels and synaptic proteins, although the mechanisms are not fully understood [7]. These findings explain, at least in part, the Meyer–Overton hypothesis, which suggests a positive relationship between the anesthetic effect and lipid solubility of organic chemicals. Similarly, it has been reported that lipid solubility is linearly correlated with the adverse effects on growth in microorganisms, such as yeast [8]. Thus, lipid solubility increases the general toxicity of organic chemicals across multiple cell types and species.

The nematode *Caenorhabditis elegans* has been widely used as a powerful model organism in basic science research, including genetics, molecular biology, and neurobiology. Moreover, its use in toxicology research has increased in recent years [9]. Several studies have shown similarities between *C. elegans* and other organisms exhibiting chemical toxicities [10]. Biological targets appear to be relatively similar between *C. elegans* and higher organisms based on the conservation of genes that compose the cellular structure, molecular signaling, and enzymes that produce biological materials [9]. Furthermore, the small body size of approximately 1 mm in adults and the fast life cycle of approximately 3 days from egg to mature adult under optimal conditions enable high-throughput screening, such as chemical screening, using large quantities of nematodes [11]. Therefore, *C. elegans* are ideal animal models for understanding chemical toxicities in organisms and their underlying cellular and molecular mechanisms. Although a standardized protocol for assessing environmental toxicity has been developed [12], no standardized protocol exists for determining chemical toxicity in humans using *C. elegans*.

The behavioral effects of anesthetics, such as halogenated hydrocarbons and halogenated ethers, and alcohols like ethanol, have been well studied in *C. elegans* [13,14]. Exposure to these anesthetics or alcohols causes changes in locomotion, such as uncoordinated and slow

movements, eventually resulting in complete immobility. It has been shown that the paralyzing effects of these volatile chemicals are highly correlated with their lipid solubility [15,16]. Genetic studies have identified several genes and molecular mechanisms underlying paralyzing effects. UNC-79 and UNC-80 encode key molecules that regulate a voltage-insensitive cation channel, the NALCN (Na+ leak channel, non-selective) channel, in *C. elegans.* Mutants of *unc-79* and *unc-80* show hypersensitive phenotypes to a subset of volatile anesthetics, while being normal or resistant to other sets of volatile anesthetics and ethanol, suggesting that UNC-79 and UNC-80 mediate responses to only a subset of volatile anesthetics and that multiple mechanisms underlie anesthetic effects in *C. elegans* [16,17]. The direct targets of volatile anesthetics and alcohols have also been identified through genetic studies of *C. elegans* combined with *in vitro* studies using their mammalian homologs [18–20]. Ethanol was shown to activate *C. elegans* and mammalian BK potassium channels *in vivo* and *in vitro*, and an alcohol-binding site of the BK channel was identified via crystallography [21,22]. It has also been revealed that several other proteins, such as ion channels and gap junction proteins localized to microdomains known as lipid rafts in the cell membrane, play important roles in the anesthetic effects of volatile chemicals in *C. elegans* [14,23].

In this study, I focused on 30 kinds of simple organic solvents with various functional groups, including alcohols, ethers, ketones, and acetate esters. These organic solvents are commonly used in chemical industries and are associated with the risk of neurotoxicity and/ or anesthetic effects from high-dose exposure in humans. The locomotion of the nematodes was comprehensively analyzed after exposure to organic solvents using a tracking system. The motility of the nematodes substantially decreased and eventually ceased after exposure to any of the 30 organic solvents. The paralyzed nematodes recovered after incubation without the organic solvents, suggesting that the organic solvents had an anesthetic effect on the nematodes. The extent of the anesthetic effects was proportional to the octanol–water partition coefficient, which reflects lipid solubility. Therefore, a Meyer–Overton relationship was observed for the adverse effects of organic solvents, which is consistent with previous reports demonstrating a positive relationship between lipid solubility and the potency of several anesthetics in *C. elegans* [15].

Furthermore, I quantified the amplitude of body bends during undulatory locomotion and found that the specific effects of organic solvents on posture during locomotion qualitatively differed depending on the solvent's functional group. The amplitude of body bending considerably increased after exposure to chemicals with short-chain ketone and acetate ester groups at concentrations lower than those that resulted in complete paralysis. Conversely, it decreased after exposure to short-chain alcohol and cellosolve groups. The increased effects on amplitude depended on TAX-4, a cyclic nucleotide-gated channel involved in sensory responses in chemosensory neurons [24], suggesting that chemosensory input is required for the increase in amplitude of body bends during locomotion. Conversely, decreased amplitude was observed in mutant backgrounds of genes required for chemosensory responses. These findings imply that organic solvents affect the neural circuit in different ways depending on the solvent's distinct functional groups.

Finally, I have demonstrated that organic solvents cause substantial defects in salt chemotaxis learning, a paradigm of learned behavior in salt chemotaxis, at concentrations lower than those affecting locomotion speed [25]. These effects did not depend on ODR-3, an alpha-subunit of the Gi/Go-like G protein that is critical for the sensory transduction of volatile compounds in chemosensory neurons [26]. These results imply that organic solvents affect the sensory neural circuits regulating salt chemotaxis plasticity in parallel with ODR-3-mediated sensory input. This study provides an observation of the multifaceted effects of toxic organic solvents on speed and posture during locomotion, as well as learned behavior in *C. elegans.*

## Materials and methods

### Cultivation of *C. elegans*

The nematode *C. elegans* were cultivated on nematode growth medium (NGM) plates using a standard protocol [27]. Wild-type N2, the *unc-29(e193)* CB193, the *tax-4(p678)* JN730, the *osm-9(ky10)* CX10, the *dyf-11(pe554)* JN554, the *odr-3(n2150)* JN1722, and the *daf-2c(pe2722)* JN2722 *C. elegans* strains were used. The *Escherichia coli* OP50 bacterial strain was used as the food source. To prepare synchronized populations of *C. elegans* for locomotion analysis, eggs were harvested from gravid adult nematodes via treatment with an alkaline hypochlorite solution. Approximately 600 eggs were cultivated on NGM plates at 18°C–20°C for 4–5 days until they reached the young adult stage.

### Recording of the nematodes' movement after exposure to organic solvents

Synchronous populations of adult *C. elegans* were transferred from the growth plates to 500 μL of $KPO_4$ buffer solution (5 mM potassium phosphate, 1 mM $CaCl_2$, 1 mM $MgSO_4$, and 0.05% gelatin). To determine the $MC_{<50}$ values, organic solvents, sodium chloride, or glucose were dissolved in $KPO_4$ buffer at 4–7 different concentrations within a range of 0.125%–6%, and the nematodes were exposed for 15, 30, 60, or 120 min. For the untreated control groups, a $KPO_4$ buffer without organic solvents was used for the exposure procedure. To examine recovery from paralysis, the nematodes were soaked in $KPO_4$ buffer containing organic solvents for 2 h at their minimum concentration that caused complete paralysis, which was defined as the average locomotion speed being below 10 μm/s, unless otherwise stated. The nematodes were then transferred into $KPO_4$ buffer without organic solvent and incubated for 30, 60, or 120 min. After these soaking procedures, a 30-μL droplet with approximately 50 nematodes was transferred onto a test plate (5 mM potassium phosphate, 1 mM $CaCl_2$, 1 mM $MgSO_4$, and 2% Bacto agar), and excess liquid was removed with paper wipers. The movement of the nematodes on the test plates was captured under a stereomicroscope (S6E, Leica Microsystems, Germany) equipped with a USB CMOS camera (J-scope, Sato Shoji Corporation, Japan) operating at 30 frames per second for approximately 1 min.

### Tracking analysis of *C. elegans* locomotion

*C. elegans* locomotion was analyzed using the WormLab software (MBF Bioscience, Vermont, USA) with 1500-frame image sequences (50-second movie at 30 frames per second). The center point of each nematode on the agar plates was tracked (Fig 1A and S1 Movie), and the movement speed of each trace was determined. The plus value represents forward movement, the minus value represents reverse movement, and an absolute value below 30 μm/s was defined as an immobile state (Fig 1B). The locomotion speed of each trace was determined as the average speed during forward movement, and the mean locomotion speed of all traces was plotted along with the standard error of the mean (SEM) on a graph after normalization with the untreated control (Fig 1D, right). Immobility was determined as the proportion of the immobile state in each trace, and the mean value of all traces along with the SEM was plotted on a graph (Fig 1D, left). The length of each nematode was determined as the length from head to tail along the central axis, and the mean length of all nematodes was plotted along with the SEM on a graph (Fig 3A). The amplitude of the body bends during locomotion was determined as the average centroid displacement, which is the distance between the midpoint and the average location of the central axis points of nematodes (Fig 5B), and the mean amplitude of each trace was plotted on violin plots (Fig 5C). At least three traces were used for the analyses.

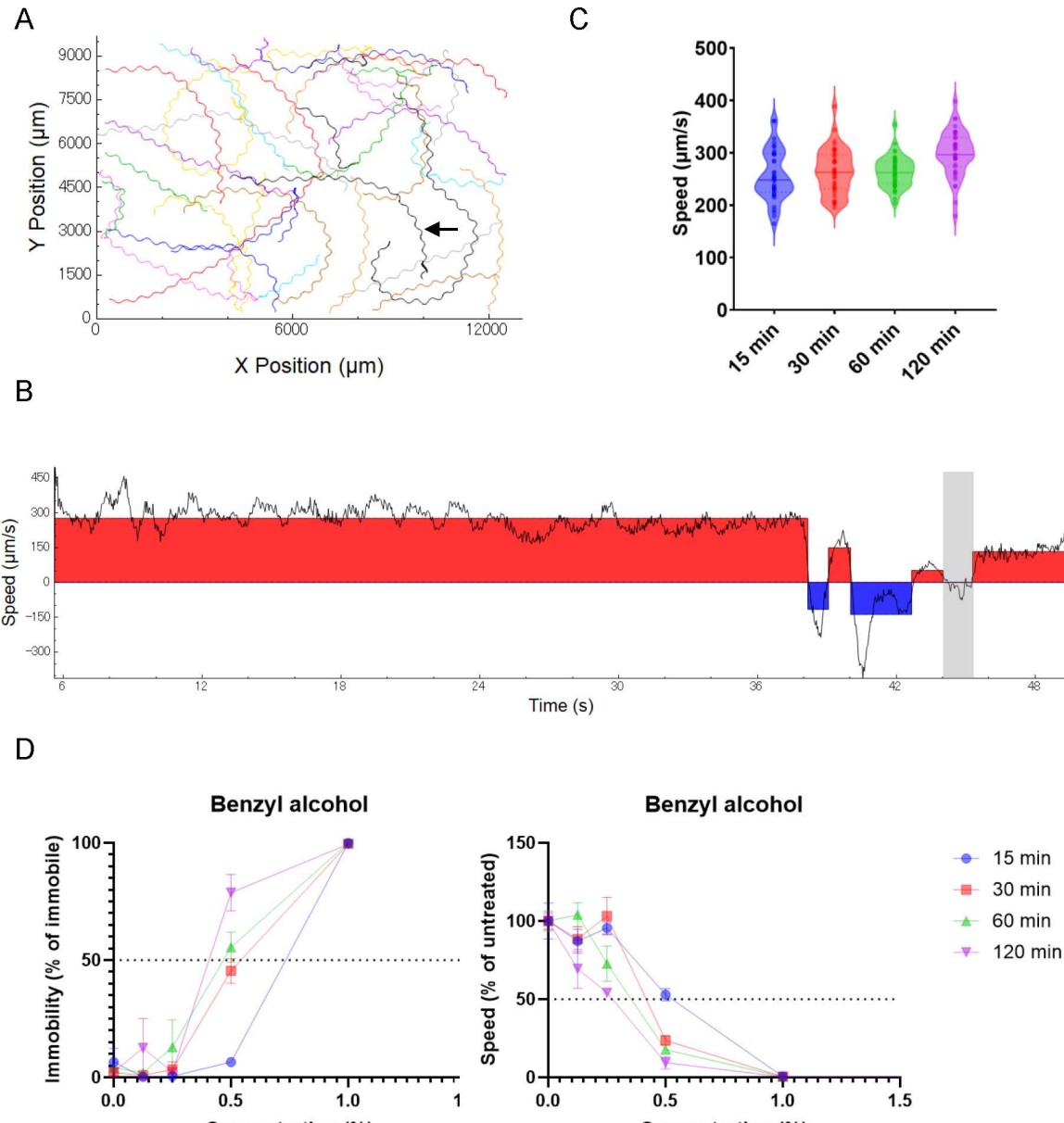

**Fig 1. Quantification of the locomotion of *C. elegans* exploring agar plates.** (A) Representative trajectories of nematode *C. elegans* moving on an agar plate within an area of approximately 1 cm² for 50 s. A trace depicted by an arrow is used for the representative analysis of the locomotion speed shown in B. (B) Locomotion speed (μm/s) during 50 s of tracking at *C. elegans*. Positive values (red region) or negative values (blue region) are defined as "forward" or "reverse" movements, respectively. Absolute values below 30 μm/s (gray region) are defined as "immobile." (C) Violin plots of locomotion speeds on an agar plate after soaking in $KPO_4$ buffer. The x-axis represents the soaking time before locomotion assay on an agar plate. Each dot represents the mean locomotion speed in each trace. The solid horizontal lines represent the median. The dotted horizontal lines represent the 25th and 75th percentiles. (D) The proportion of the immobile state (left) and locomotion speed (right) after exposure to benzyl alcohol for 15, 30, 60, or 120 min. Each data point represents the mean ± the standard error of the mean (SEM). The locomotion speed was normalized to the average value of the untreated control.

## Salt chemotaxis learning

A salt chemotaxis learning assay was performed according to a previously published procedure with some modifications [25] (S18 Fig). To prepare a chemotaxis test plate, an agar block containing NaCl (50 mM NaCl, 5 mM potassium phosphate, 1 mM $CaCl_2$, 1 mM

$MgSO_4$, and 2% Bacto agar) was placed at the edge of an agar plate (5 mM potassium phosphate, 1 mM $CaCl_2$, 1 mM $MgSO_4$, and 2% Bacto agar) to form a NaCl concentration gradient by overnight incubation at room temperature (S18A Fig, right). For conditioning, the nematodes were transferred into 500 μL of $KPO_4$ buffer solution (5 mM potassium phosphate, 1 mM $CaCl_2$, 1 mM $MgSO_4$, and 0.05% gelatin) with or without 20 mM NaCl and incubated for 1 h, and named NaCl(+) or NaCl(−) conditioning, respectively. To assess the effect of organic solvent exposure for 1 h, the organic solvents were dissolved in $KPO_4$ buffer solution for NaCl(+) or NaCl(−) conditioning. To assess the effect of organic solvent exposure for 1 min, the nematodes were soaked in buffer solutions with or without NaCl for 1 h, then transferred into NaCl(+) or NaCl(−) buffer, respectively, which included the organic solvents, and incubated for 1 min. After conditioning, a droplet with approximately 50–100 nematodes was transferred to the chemotaxis test plate, and excess liquid was removed with paper wipes. Just before transferring the nematodes onto the chemotaxis test plate, the agar block was removed, and 0.5 μL of 1 M sodium azide was spotted at both positions where the agar block was placed and the opposite side to anesthetize the nematodes at those positions (S18B Fig). After the nematodes freely explored the test plate for 15 min, the numbers of anesthetized nematodes at each position were counted, and the chemotaxis index was calculated, as shown in S18B Fig. When the chemotaxis index value is 1.0, all nematodes that migrated from the starting area moved toward the high-NaCl area. When the chemotaxis index value is −1.0, all nematodes that migrated from the starting area moved toward the low-NaCl area. Assays were repeated six times.

## Calculation of internal alcohol concentrations

Internal ethanol concentrations were calculated according to a previous report with some modifications [28,29]. Synchronized populations of approximately 500 wild-type *C. elegans* at the young adult stage were transferred to 500 μL of $KPO_4$ buffer solution (5 mM potassium phosphate, 1 mM $CaCl_2$, 1 mM $MgSO_4$, and 0.05% gelatin) with 500 mM ethanol and incubated for 10 or 50 min at room temperature. The nematodes were washed twice with ice-cold $ddH_2O$, and the supernatants were removed. The volume of the nematode-containing solution was adjusted to approximately 25 μL. After freezing at −80°C for >30 min, the nematodes were thawed and ground with a pestle in the tube on ice. The alcohol concentration in the homogenate solution was determined according to the manufacturer's instructions using an Alcohol Assay Kit (Cell Biolabs, USA). The volume of *C. elegans* was calculated by assuming that a single nematode is a cylinder [28]. Based on movies for locomotion analysis, the length and width of a nematode were determined as the length of the central axis and the average length of cross-sections over the entire body, respectively, using WormLab (S19 Fig). The average body length and half of the average body width were used as the height (h) and radius (d) of the nematode, respectively, and the volume was calculated as a cylinder according to the following equation: volume = $\pi r^2 h$. The internal ethanol concentration ($C_2$) of the nematodes was calculated using the following equation: $C_1 V_1 = C_2 V_2$, where $C_1$ = ethanol concentration of the homogenate solution, $V_1$ = total volume of the homogenate solution, and $V_2$ = the estimated volume of 500 nematodes. Three biological replicates were performed for each condition.

## Statistics and raw data

Statistical analyses were performed using GraphPad Prism 10. A simple linear regression analysis was performed to calculate the values of the coefficient of determination, $R^2$ (Figs 2A–2D,

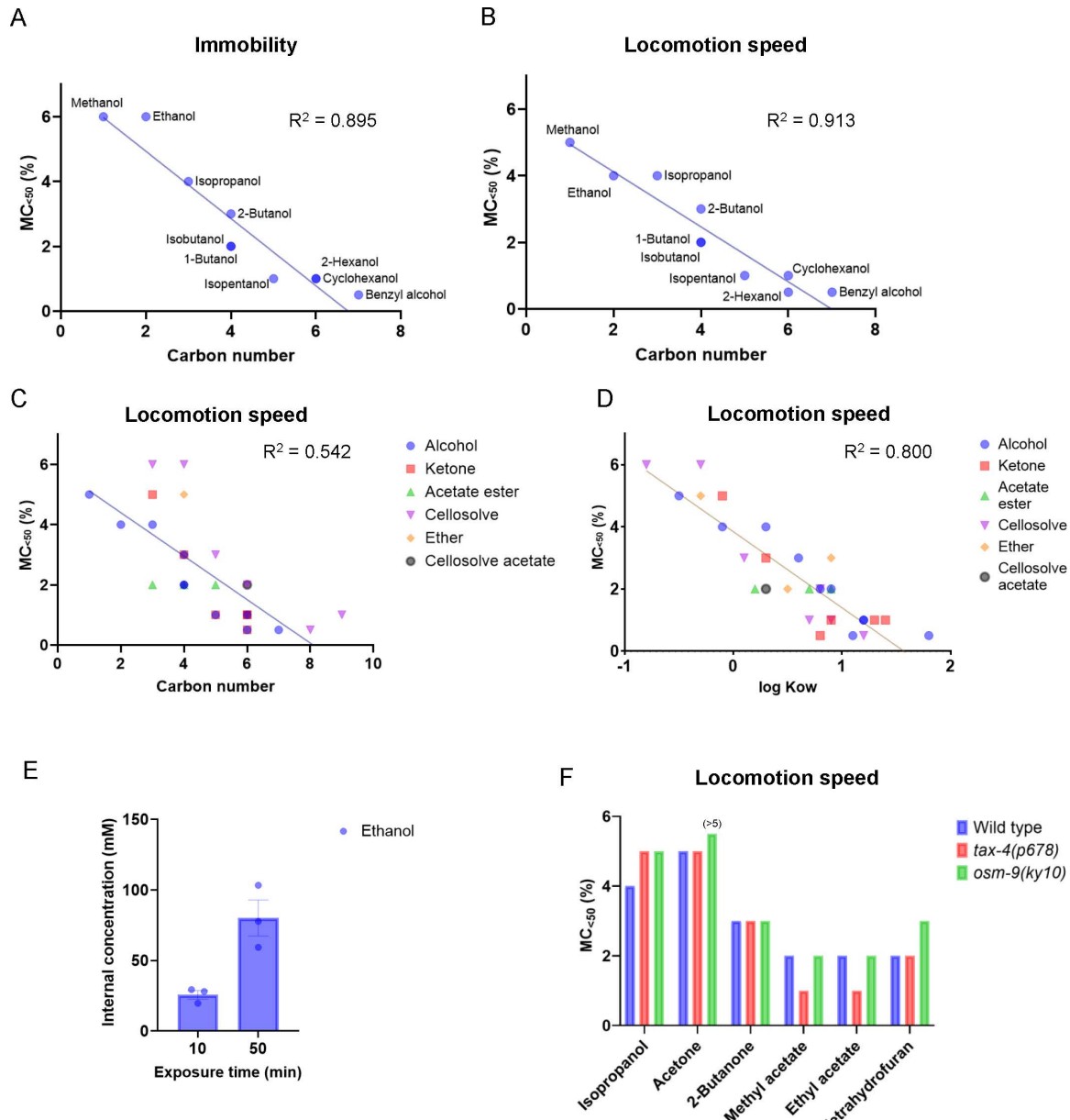

**Fig 2. Correlation between acute behavioral toxicity with the organic solvent's carbon number and lipid solubility.** (A, B) Relationship between behavioral toxicity ($MC_{<50}$) after alcohol exposure for 1 h and the carbon number of alcohols. The proportion of the immobile state (A) and locomotion speed (B) are used as the endpoints. (C, D) Relationship between the $MC_{<50}$ after exposure for 1 h to organic solvents and the chemical's carbon numbers (C) or octanol–water partition coefficients (D). The locomotion speed is used as the endpoint. $R^2$ values were determined using simple linear regression analysis. (E) Estimated internal concentrations of ethanol after exposure to 500 mM ethanol for 10 or 50 min. Each bar represents the mean ± the standard error of the mean (SEM). n = 3 biological replicates. (F) Comparison of the $MC_{<50}$ after exposure for 1 h to organic solvents among the wild type, *tax-4(p678)*, and *osm-9(ky10)* nematodes. The locomotion speed is used as the endpoint. The $MC_{<50}$ after acetone exposure in *osm-9(ky10)* is above 5% and is indicated as (>5) above the bar.

7C and S4–S6). For multiple comparison tests, an unpaired *t*-test with Holm–Sidak correction (Figs 3B, 5D and S9, S11C, S17), a one-way ANOVA with Dunnett's test (Figs 5C, 6, 7D and S13, S15), and a two-way ANOVA with Dunnett's test (Figs 7B, 8 and S14, S16) were used. Raw data and statistics are provided in S1–S4 Files.

## Results

### Behavioral toxicity of monohydric alcohols is proportional to the carbon number of the alcohol

To assess the effects of exposure to organic solvents, the solvents were dissolved in $KPO_4$ buffer solution, and *C. elegans* were soaked in buffer for 15, 30, 60, or 120 min (Materials and Methods). After exposure, *C. elegans* were transferred to agar plates, and locomotion was quantified using a tracking system to evaluate the effects of organic solvents on behavior. The speed of freely moving nematodes on an agar plate was measured, and the behavioral states were determined: forward movement, reverse movement, and an immobile state were determined based on the speed and direction during locomotion (Fig 1A–1C and S1 Movie). First, the effects of exposure to 10 kinds of monohydric alcohols with carbon numbers ranging from C1 to C7 were tested. The speed during forward movement and the proportion of the immobile state were quantified after exposure to alcohols at several concentrations for 15–120 min (Figs 1D and S1). For example, after exposure to 0.5% benzyl alcohol, the proportions of the immobile state and locomotion speed substantially increased and decreased, respectively. The proportion of the immobile state increased to >50% after exposure for >1 h, and the speed decreased to <50% after exposure for >30 min (Fig 1D and S2 Movie). The minimum concentrations of each alcohol that increased the immobility state to >50% or decreased the speed of locomotion to <50% after chemical exposure were determined, and the concentration was defined as $MC_{<50}$. The $MC_{<50}$ was established for the 10 monohydric alcohols (Figs 1D and S1). It has been reported that the inhibitory effects of alcohols on growth increase with increasing numbers of carbon atoms in the yeast *Saccharomyces cerevisiae* [8]. The $MC_{<50}$ for *C. elegans* locomotion was compared with the carbon numbers of the alcohols to which they were exposed. The $MC_{<50}$ for immobility and locomotion speed exhibited a strong correlation with the carbon numbers of the alcohols (Fig 2A and 2B). These results suggest that the behavioral toxicity of monohydric alcohol can be determined by immobility or locomotion speed and is proportional to its carbon number in *C. elegans*. To quantify the behavioral toxicity of chemicals, locomotion speed was primarily used as an endpoint in subsequent analyses.

### Behavioral toxicity of organic solvents is proportional to the chemical's lipid solubility

Next, the behavioral toxicity of organic solvents with distinct functional groups (ketone, acetate ester, ether, cellosolve, and cellosolve acetate) was determined using locomotion speed as an endpoint (S2 and S3 Figs). Similar to the adverse effects of alcohol, exposure to these organic solvents decreased locomotion speed on an agar plate (S2 and S3 Figs). Furthermore, the toxicities of chemicals with ketone or cellosolve groups were proportional to the chemical's carbon numbers (S4 Fig). However, compared to the correlation of the $MC_{<50}$ with the carbon number of the 10 kinds of alcohols (Fig 2B), the correlation for the 30 kinds of organic solvents was low, presumably because the distinct functional groups exert various effects on locomotion (Fig 2C). The $MC_{<50}$ was then compared with the lipid solubility of organic solvents, which is an important property for several biological processes, such as skin permeability and anesthetic effect. High correlation levels were observed when the $MC_{<50}$ was compared with the octanol–water partition coefficient, which reflects lipid solubility (Figs 2D and S5). Meanwhile, no significant correlation was observed between the $MC_{<50}$ and the chemical melting points, which affect skin permeability (S6 Fig). These results suggest that lipid solubility is an important property of organic solvent toxicity on locomotion in *C. elegans*.

According to a published method with some modifications, the concentration of ethanol in *C. elegans* internal tissues was estimated [28]. The estimated internal concentrations of

ethanol after exposure to 500 mM ethanol in $KPO_4$ buffer solution for 10 or 50 min were 25.7 ± 3.03 mM or 80.1 ± 12.7 mM, respectively (Fig 2E). The internal ethanol concentration after 50 min exposure was comparable to that in the previous report, where ethanol was exposed to the nematodes on an agar plate (89.3 ± 8.8 mM) [28]. Meanwhile, the internal ethanol concentration after 10 min exposure was lower than that in the previous report (67.5 ± 7.1 mM) [28]. The difference between the exposure methods (exposure on a plate vs. in liquid) may reflect differences in the time course of ethanol penetration between the current and previous studies [28].

## Low-molecular-weight ketones and ethers cause body shrinkage followed by relaxation

Levamisole, an anthelmintic agent, causes paralysis in *C. elegans*. Levamisole acts as a potent agonist of acetylcholine receptors, and its exposure causes muscle contraction followed by muscle relaxation [30]. I confirmed that levamisole reduced locomotion speed and eventually ceased movement at 100 μM after 1 h of exposure (S7A and S7B Fig). Impaired locomotion was associated with body shrinkage (S7C and S7D Fig, middle). Most nematodes were completely paralyzed by levamisole exposure at 1,000 μM, and body relaxation was observed after 1 h of exposure (S7A–S7D Fig, right). Similar to these observations, exposure to a ketone, 2-butanone, or an ether, tetrahydrofuran, gradually decreased body length at effective concentrations for impaired locomotion, followed by relaxation at concentrations that led to complete paralysis (Figs 3A and S2A, S2C, S8). The other C3–C6 ketones and C4 ethers, except 2-hexanone, exhibited similar effects (Fig 3B). In contrast, most alcohol, acetate ester, cellosolve, and cellosolve acetate chemicals had no significant effect on body shrinkage, except 2-hexanol and methyl acetate (S9 Fig).

To assess whether 2-butanone and tetrahydrofuran affect muscle function via levamisole-sensitive acetylcholine receptors, the effects of exposure to these chemicals were examined in a loss-of-function mutant of UNC-29, an essential subunit of levamisole-sensitive acetylcholine receptors expressed in the body-wall muscle [31]. It was first confirmed that the loss-of-function *unc-29(e193)* mutant was resistant to levamisole exposure, which is consistent with a previous report that the acetylcholine receptors containing the UNC-29 subunit are the primary target of levamisole in the muscle (S10A Fig). In contrast, exposure to 2-butanone or tetrahydrofuran caused substantial locomotion impairment in the *unc-29* mutant (S10B and S10C Fig). The *unc-29* mutant was sensitive to organic solvents, including 2-butanone and tetrahydrofuran, when immobility and locomotion speed were used as endpoints (S11A and S11B Fig). Exposure to acetone, 2-butanone, or tetrahydrofuran also caused a significant reduction in body size in the *unc-29* mutant (S11C Fig). These results suggest that low-molecular-weight ketones and ethers reduce locomotion speed and body size in a UNC-29 independent manner.

## Recovery from paralysis caused by exposure to organic solvents

To examine whether locomotion impairment after exposure to organic solvent is reversible, the locomotion of nematodes was assessed after recovery in buffer solution without organic solvent. Nematodes were exposed to organic solvents for 2 h at concentrations that caused complete paralysis and were then soaked in $KPO_4$ buffer solution without the solvents. Impaired locomotion was recovered within 2 h of soaking in buffer after paralysis caused by exposure to most organic solvents, except benzyl alcohol, 2-pentanone, 2-hexanone, and ethyl acetate (Fig 4). Complete paralysis after exposure to benzyl alcohol, 2-pentanone, 2-hexanone, and ethyl acetate for shorter periods (15 or 30 min) was recovered by soaking in a

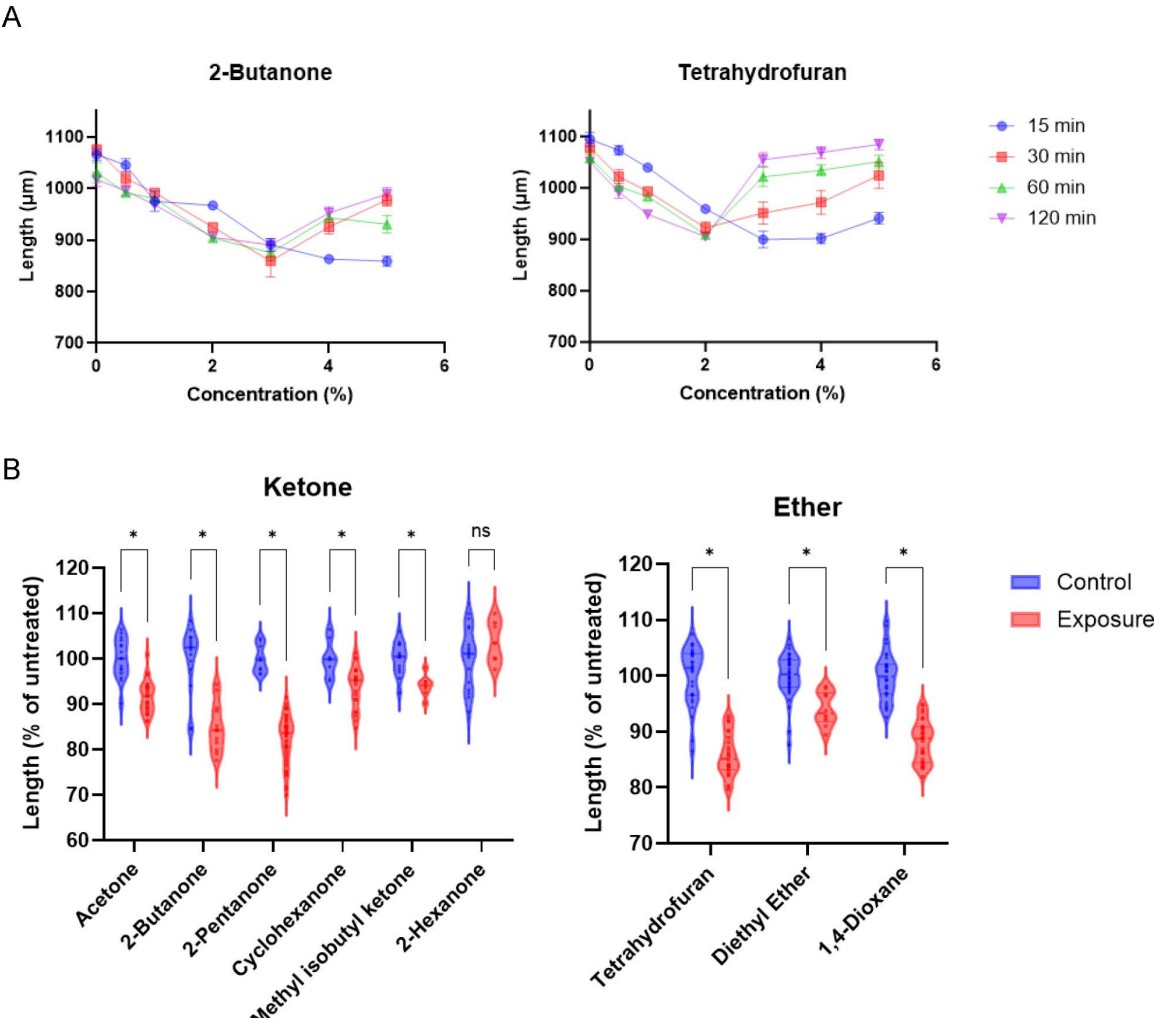

**Fig 3. Exposure to ketone and ether causes body shrinkage followed by relaxation.** (A) Changes in body length after exposure to 2-butanone (left) or tetrahydrofuran (right). Each data point represents the mean ± the standard error of the mean (SEM). (B) Violin plots of the body lengths of nematodes after exposure to ketones (left) or ethers (right). The exposure concentrations are the $MC_{<50}$ values determined by locomotion speed after 1 h of exposure. Data were normalized to the average values of the control. Each dot represents the body lengths of a nematode after exposure (red) or without exposure (blue). *$P < 0.05$, unpaired *t*-test with Holm–Sidak correction.

buffer without the solvents (S12 Fig). These results suggest that nematodes are paralyzed after short-term exposure to organic solvents, which is reversible, whereas long-term exposure to some organic solvents can cause irreversible locomotion impairment. These phenotypes are reminiscent of those observed after exposure to human anesthetics in *C. elegans* [15].

## Organic solvents have qualitatively different effects on body bending amplitude during locomotion

In addition to the general adverse effects of organic solvents on locomotion, is there a specific effect that is qualitatively different based on the solvent's chemical characteristics, such as its functional group (Fig 5A)? It has been reported that ethanol exposure decreases the amplitude of body bending and speed during locomotion in *C. elegans* [20]; therefore, the amplitude of

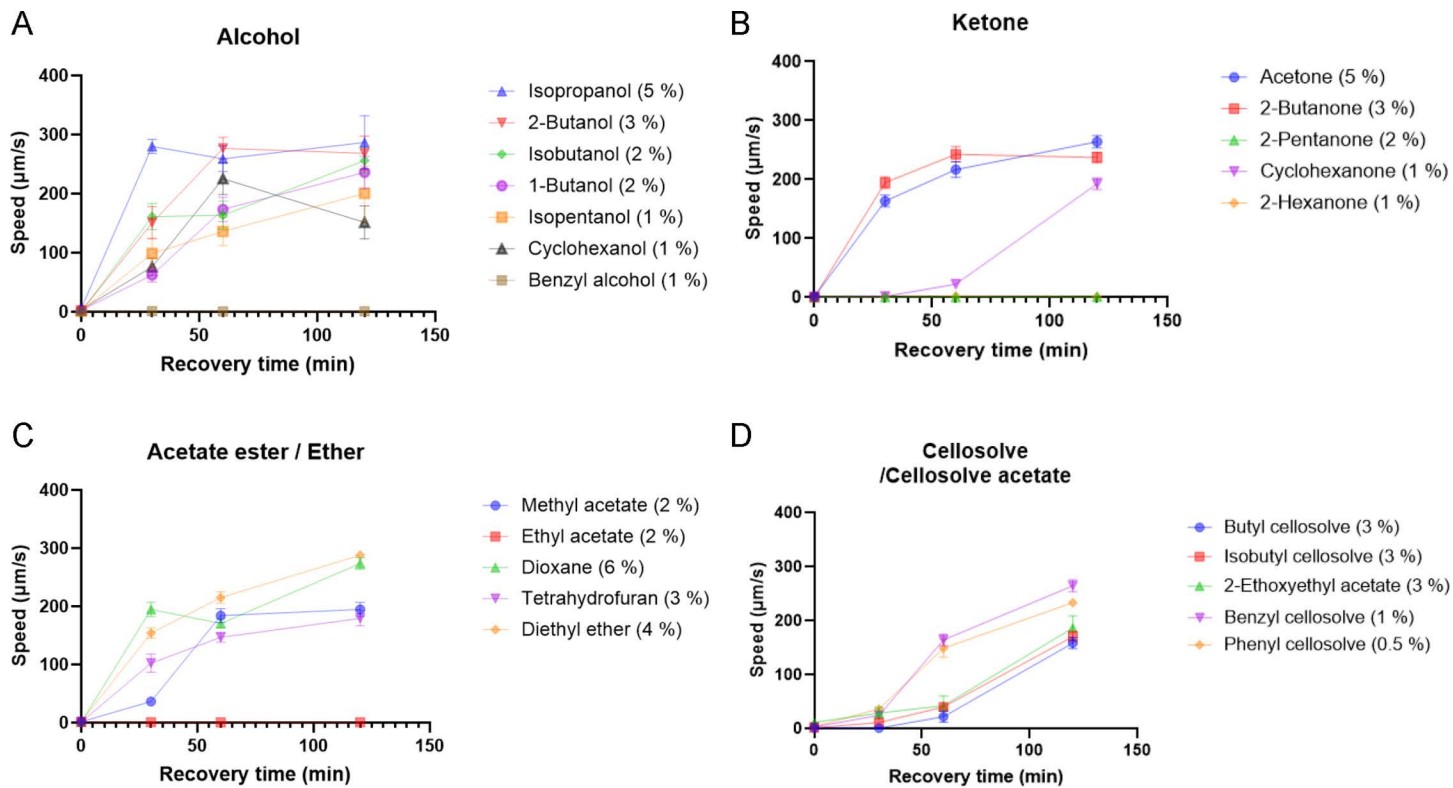

**Fig 4. Time course of recovery from paralysis after exposure to organic solvents.** (A–D) Locomotion speed after recovery in $KOP_4$ buffer solution after 2 h of exposure to organic solvent. The exposure concentrations of the organic solvents are the minimum concentrations that cause complete paralysis. Each data point represents the mean ± the standard error of the mean (SEM).

body bending during locomotion was quantified (Fig 5B). As reported for ethanol exposure, exposure to alcohols, including methanol, isopropanol, and 2-butanol, for 15 min decreased the amplitude of body bends during locomotion at concentrations lower than $MC_{<50}$ (Figs 5C and S13A and S3 Movie). Similar effects on body bends were observed after exposure to cellosolves, including methyl, ethyl, and isopropyl cellosolve, as well as cellosolve acetate; these organic solvents decreased the amplitude of body bends (S13B and S13E Fig).

Interestingly, when nematodes were exposed to ketones and acetate esters, the effects on body bending were opposite to those observed with alcohols and cellosolve. Exposure to ketones, including acetone, 2-butanone, and 2-pentanone, and acetate esters, including methyl acetate, ethyl acetate, and isopropyl acetate, for 15 min significantly increased the amplitude of body bends during locomotion at concentrations lower than $MC_{<50}$ (Figs 5C and S13C and S4 Movie). The effects of ether on amplitude were intricate: 1,4-dioxane decreased the amplitude of body bends; diethyl ether did not affect body bends; and tetrahydrofuran decreased and increased the amplitude of body bends at concentrations of 0.5% and 2%, respectively (Figs 5C and S13D and S5 Movie). These results suggest that exposure to organic solvents affects posture during locomotion differently at concentrations lower than those that cause cessation of motility.

### Involvement of chemosensory functions in altered locomotion speed and posture after exposure to organic solvent

*C. elegans* receive external volatile chemicals, such as alcohols, ketones, and ethers, through their chemosensory neurons and respond to these chemicals with attraction or repulsion

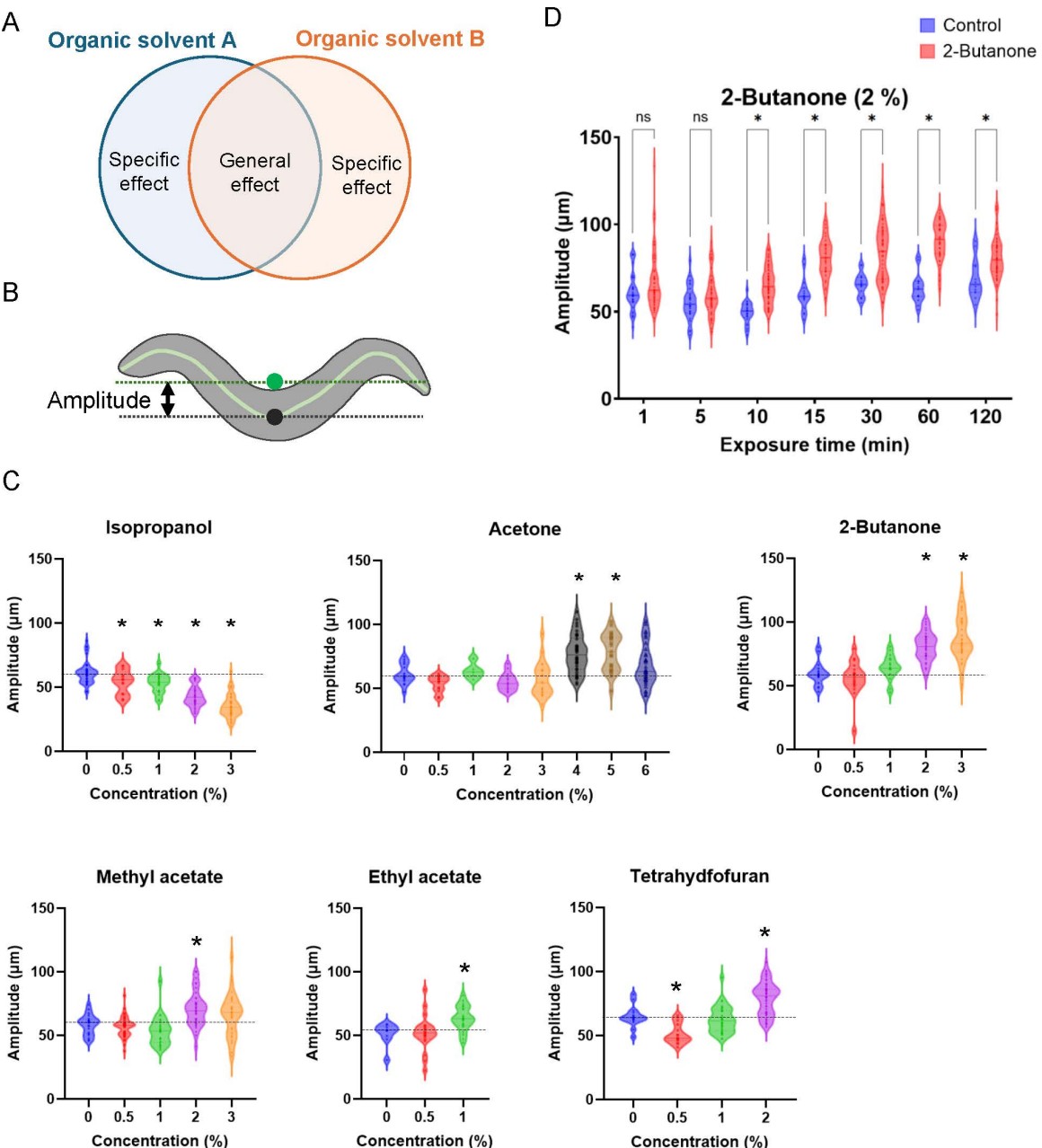

**Fig 5. Alcohols and ketones affect posture during locomotion in opposite directions.** (A) Working model of the toxicity of organic solvents. Organic solvents have general toxicities that induce similar effects on organisms, as well as specific toxic effects that are qualitatively different among chemicals. (B) Quantification of the amplitude of body bending during locomotion. The midpoint (black dot) and the average location of the central axis points (green dot) are shown. (C) Violin amplitude plots after soaking in a buffer containing organic solvents for 15 min. Each dot represents the mean amplitude during each track. *$P < 0.05$, one-way ANOVA with Dunnett's test, compared with the no-exposure control. (D) Violin amplitude plots after soaking in a buffer with (red) or without (blue) 2% 2-butanone for the indicated time. *$P < 0.05$, unpaired *t*-test with Holm–Sidak correction.

behaviors, known as chemotaxis [32]. The sensory transduction of chemosensory cues is primarily mediated by ion channel effectors, TAX-4/TAX-2, a cyclic nucleotide-dependent cation channel complex, and OSM-9/OCR-2, a transient receptor potential channel superfamily protein complex [24,33,34]. To explore the extent to which chemosensory input is required

for behavioral changes after prolonged exposure to organic solvents, locomotion speed was examined in *tax-4* and *osm-9* mutants after exposure to several organic solvents (Figs 2F and S14). Both the *tax-4* and *osm-9* mutants showed increased $MC_{<50}$ values for isopropanol, that is, they are resistant to a decrease in locomotion speed after isopropanol exposure. Only the *osm-9* mutant exhibited resistance to a decrease in locomotion speed after exposure to acetone and tetrahydrofuran. In contrast, the *tax-4* mutant showed an enhanced decrease in locomotion speed after exposure to methyl acetate and ethyl acetate. Both mutant strains showed a decrease in locomotion speed after exposure to 2-butanone, comparable to that of the wild type. These results suggest that the chemosensory input of organic solvents may influence locomotion speed after prolonged exposure, with the extent and direction of the effects varying depending on the solvent.

Next, changes in the amplitude of body bending after exposure to organic solvent were examined in *tax-4* and *osm-9* mutants. The *osm-9* mutant showed no substantial difference in increased amplitude after 15 min of exposure to 2-butanone, methyl acetate, ethyl acetate, and tetrahydrofuran, and a decrease in amplitude after isopropanol exposure compared with the wild type (Figs 5C and S15). Meanwhile, the *tax-4* mutant showed defects in increased amplitude after 15 min of exposure to acetone, 2-butanone, methyl acetate, ethyl acetate, or tetrahydrofuran, but showed a normal decrease in amplitude after isopropanol exposure (Figs 5C and 6). These results suggest that TAX-4-mediated sensory input is required for the increased

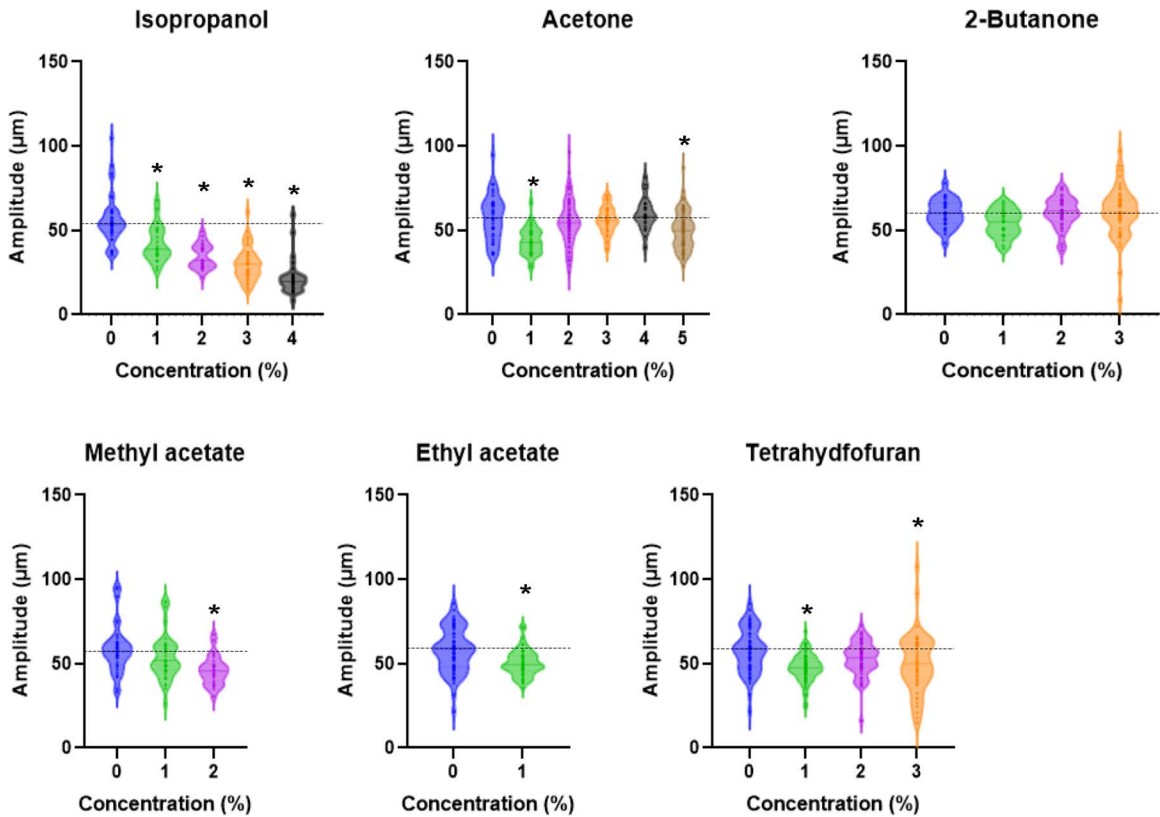

**Fig 6. The *tax-4* mutant exhibits defects in increased body bending amplitude during locomotion after exposure to ketones, acetate esters, and tetrahydrofuran.** Violin plots showing the amplitudes after soaking in a buffer containing organic solvents for 15 min in *tax-4(p678)* nematodes. Each dot represents the mean amplitude during each track. *$P < 0.05$, one-way ANOVA with Dunnett's test or Welch's t-test (for Ethyl acetate), compared with the no-exposure control.

amplitude of body bending after exposure to organic solvents such as ketones and acetate esters. It has been shown that *C. elegans* respond to decreased 2-butanone concentrations to promote 2-butanone attraction, where the AWC chemosensory neurons sense rapid changes in external 2-butanone concentrations [35]. To explore whether the increased amplitude of body bending after 2-butanone exposure is induced by rapid changes in 2-butanone concentrations, the amplitude of body bending was examined after 1, 5, or 10 min of exposure to 2% 2-butanone in the wild type. An increased amplitude of body bending was not observed after 1- or 5-min exposure to 2-butanone; however, 10-min exposure to 2-butanone significantly increased the amplitude (Fig 5D). This suggests that prolonged exposure (>5 min) is required for long-lasting increase in the bending amplitude.

## Toxicity of organic solvents is not explained merely by dehydration due to high osmotic pressure

*C. elegans* locomotion is affected by dehydration in hyperosmotic environments. Previous reports have shown that high concentrations of less toxic water-soluble chemicals, such as salt and sugar, exposed to *C. elegans* result in changes in phenotypes, such as motility and body length, which are used to study the effects of external osmotic pressure [36,37]. In this study, to examine the effects of osmotic pressure on locomotion, NaCl or glucose was dissolved in ~15 mOsm $KPO_4$ buffer (also used to dissolve organic solvents in this study), and locomotion speed and body length were quantified after exposure to the buffer solution. Exposure to a >515 mOsm solution containing >250 mM NaCl for 15 min decreased locomotion speed by >50%, and a subsequent 15-min exposure (for a total of 30 min exposure) further decreased locomotion speed (S16A Fig). The decrease in locomotion speed is not primarily due to the chemosensory response, as similar changes were observed in a *dyf-11* mutant in which the chemosensory response to water-soluble chemicals is abolished [38] (S16B Fig). As previously reported [36], decreased motility was recovered after subsequent exposure to the same solution for 30 or 90 min (total exposure of 60 or 120 min) (S16 Fig).

Exposure to >515 mOsm solution containing >499 mM (9%) glucose for 15 min also decreased locomotion speed by >50%, and the decreased motility was recovered after subsequent 105 min of exposure (total of 120 min) in the same solution (Figs 7A and S17A). The $MC_{<50}$ (mM) value of glucose was 499 mM (9%) after exposure for 15 min. The $MC_{<50}$ (mM) values for organic solvents with high hydrophilicity, such as methanol, ethanol, acetone, and 1,4-dioxane, were higher than that of glucose (Fig 7C, left): 978 mM (4%) methanol, 685 mM (4%) ethanol, 675 mM (5%) acetone, and 702 mM (6%) 1,4-dioxane, when exposed for 15 min. Organic solvents with lower hydrophilicity were more toxic and showed lower $MC_{<50}$ (mM) values than glucose (Fig 7C, left). For example, the $MC_{<50}$ (mM) value after 15 min of benzyl alcohol exposure was 96 mM (1%). As confirmed by the $MC_{<50}$ values expressed as volume percent (Figs 2D and S5), the $MC_{<50}$ (mM) values correlated with the octanol–water partition coefficient for 30 organic solvents (Fig 7C). Unlike NaCl and glucose, decreased motility after 15 min of exposure to each of the 28 organic solvents was not recovered by subsequent 105 min of exposure (total of 120 min) in the same solution, implying that organic solvents cause adverse effects on motility through mechanisms partly different from those caused by hyperosmolarity from NaCl or glucose exposure (Figs 7C right and S17). The two exceptions were exposure to methanol and methyl cellosolve, in which reduced motility after 15 min of exposure was significantly recovered after 105 min of subsequent exposure in the same buffer (S17B and S17F Fig). This suggests that reduced motility after exposure to organic solvents with high hydrophilicity, such as methanol and methyl cellosolve, may occur, at least in part due to dehydration from hyperosmotic pressure.

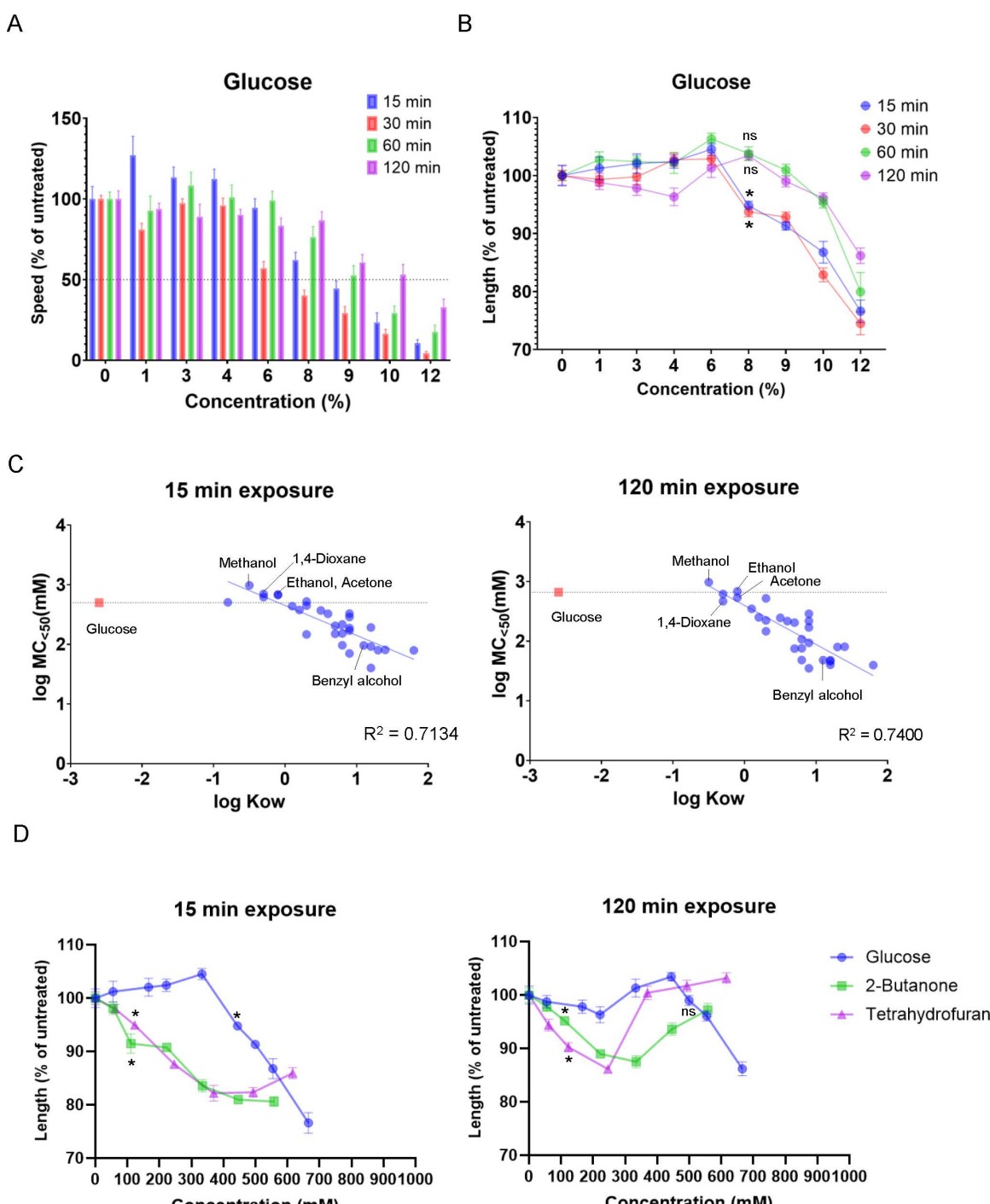

**Fig 7. Effects of glucose exposure on motility and body length, and comparison among the effects of glucose and organic solvent exposure on motility.** (A, B) Locomotion speed (A) and body length (B) after glucose exposure for 15, 30, 60, or 120 min. Data were normalized to the average value of the untreated control. (C) Relationship between the logarithm of $MC_{<50}$ (mM) and the logarithm of octanol–water partition coefficients (Kow). $R^2$ values were determined via simple linear regression analysis using values for organic solvents (blue dots). (D) Comparison of body length after glucose, 2-butanone, and tetrahydrofuran exposure. Each bar (A) or data point (B, D) represents the mean ± standard error of the mean (SEM). *$P < 0.05$, one-way ANOVA with Dunnett's test, compared with the non-exposure control (B, D).

As previously reported [37], body length significantly decreases in hyperosmotic environments. After exposure to >415 mOsm solution containing >200 mM NaCl for 15 min, body length significantly decreased, but subsequent exposure for 105 min (total of 120 min) in the same buffer recovered body length (S16 Fig). Similarly, exposure to >460 mOsm solution containing >444 mM (8%) glucose for 15 min resulted in significant body shrinkage, but subsequent exposure recovered body length (Fig 7B). Meanwhile, organic solvent-induced shrinkage occurred at lower concentrations than that caused by glucose exposure, and body shrinkage was not recovered by subsequent exposure. Exposure to 111 mM (1%) 2-butanone or 123 mM (1%) tetrahydrofuran for 15 or 120 min induced significant body shrinkage (Fig 7D). These data suggest that organic solvents, such as 2-butanone and tetrahydrofuran, cause body shrinkage through mechanisms partly different from those of dehydration due to high osmotic pressure.

## Organic solvents cause deficits in salt chemotaxis learning at concentrations lower than those that affect locomotion

To explore whether organic solvent exposure affects sensory processing in *C. elegans*, salt chemotaxis learning, a paradigm of behavioral plasticity, was examined [25]. *C. elegans* respond to various environmental chemicals, including NaCl, and exhibit behavioral plasticity appropriate for the given situation [39]. Nematodes are attracted toward NaCl, whereas after prolonged exposure to high concentrations of NaCl under starvation conditions, they avoid NaCl. This behavioral plasticity, termed "salt chemotaxis learning," is interpreted as a form of learned behavior in which nematodes memorize NaCl concentrations in their environments experienced during starvation and avoid environments without food, using the NaCl concentration as a repulsive cue [40]. Organic solvents were added to a buffer for conditioning, in which the nematodes were exposed to the organic solvents in the presence or absence of NaCl for 1 h to evaluate the effects of the organic solvents on NaCl avoidance or attraction, respectively (S18 Fig). Organic solvents were added at concentrations that had no significant effect on locomotion speed on an agar plate (S1–S3 Figs). The nematodes were attracted to NaCl after exposure to organic solvents in the absence of NaCl, except for a significant defect observed when exposed to ethyl acetate (Fig 8A). Therefore, the NaCl sensation was not largely affected by exposure to organic solvents at doses that did not cause any significant defects in locomotion speed.

Meanwhile, significant defects were observed in NaCl avoidance after exposure in the presence of NaCl when exposed to any of the organic solvents tested, including alcohol, ketone, acetate ester, ether, and cellosolve, except for methyl cellosolve (Fig 8A). These results suggest that these organic solvents cause defects in salt chemotaxis learning at concentrations lower than those that affect locomotion speed. Substantial defects in salt chemotaxis learning were not observed with exposure to any of the organic solvents for 1 min after starvation conditioning in the presence of NaCl, suggesting that prolonged exposure to organic solvents during conditioning is required for the establishment of suppressive effects on salt chemotaxis learning (Fig 8B).

In wild environments, *C. elegans* are commonly found in microbe-rich, rotting plant materials, where nematodes can be exposed to various organic compounds produced by microbes [41]. Sensory inputs from organic compounds may influence diverse behaviors, including salt chemotaxis learning. To test the extent to which sensory inputs from organic solvents affect salt chemotaxis learning, mutants of ODR-3, an alpha-subunit of the Gi/Go-like G protein expressed in chemosensory neurons, were examined. *odr-3* mutants are highly defective in most responses mediated by the AWA, AWB, AWC, and ASH neurons, which are the major

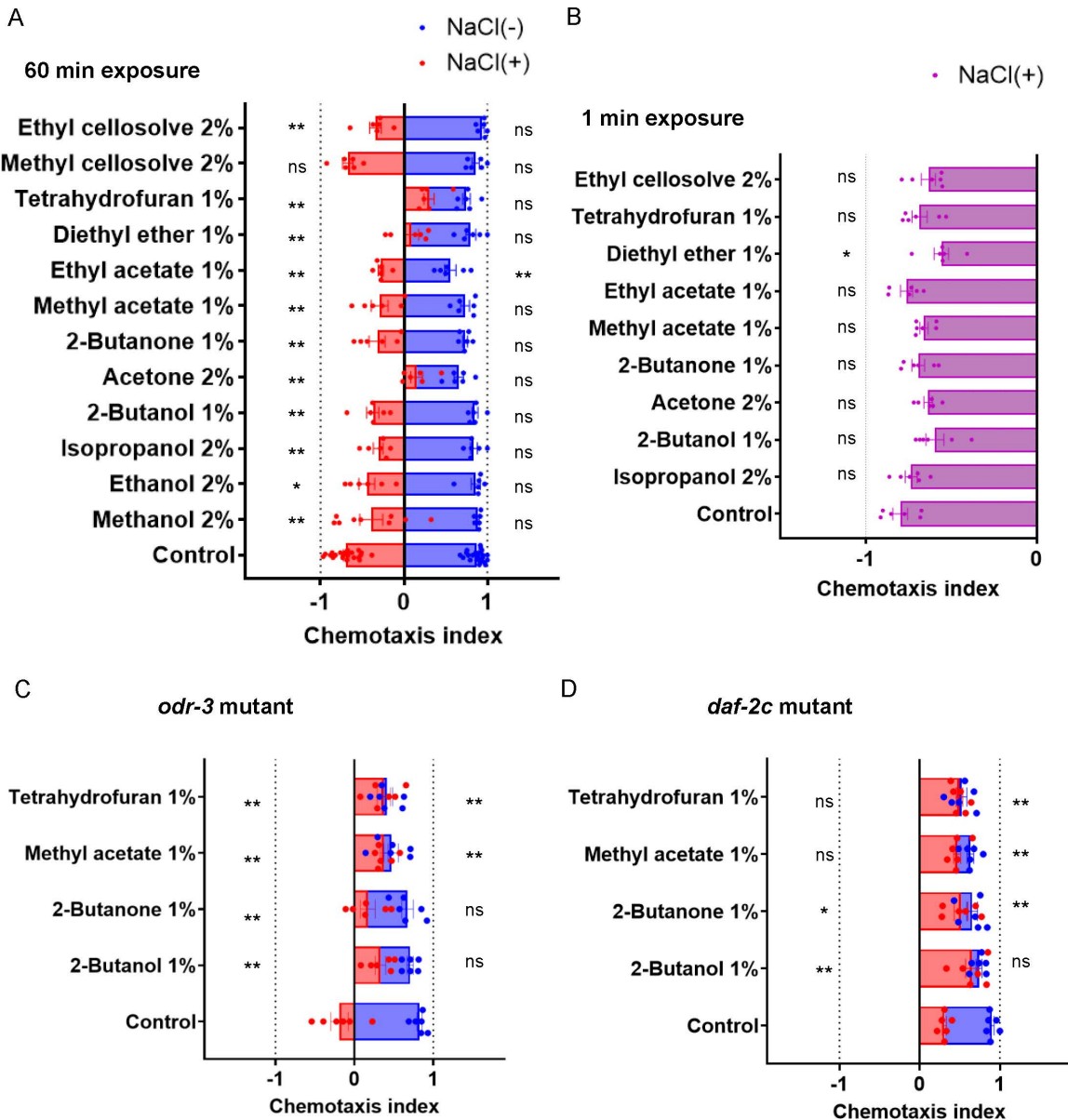

**Fig 8. Organic solvent exposure causes defects in salt chemotaxis learning.** Salt chemotaxis after soaking in buffer with or without NaCl. Nematodes were exposed to organic solvents for 1 h during the soaking procedure (A, C, D) or for 1 min after the soaking procedure (B). The exposure concentrations of organic solvents are the maximum concentrations at which no significant defect was observed in locomotion speed after exposure for 1 h. The wild type (A, B), the *odr-3(n2150)* mutant (C), and the *daf-2c(pe2722)* mutant (D) were used. The chemotaxis index represents the extent and direction of chemotaxis, where +1 and −1 indicate that all nematodes are attracted to or avoid NaCl, respectively. Each dot represents the chemotaxis index calculated for each trial (n = 6 assays). The bar and error bars indicate the mean ± the standard error of the mean (SEM). *P < 0.05; **P < 0.01, two-way ANOVA with Dunnett's test, compared with the no-exposure control.

chemosensory neurons responsible for chemotaxis to volatile compounds [26]. As suggested in previous reports [42], the *odr-3* mutant was defective in NaCl avoidance after NaCl conditioning under starvation conditions (Fig 8C). Importantly, exposure to the organic solvents 2-butanol, 2-butanone, methyl acetate, and tetrahydrofuran caused significant defects in salt

chemotaxis learning in the *odr-3* mutant, suggesting that sensory input mediated by ODR-3 is not required for the suppression of salt chemotaxis learning by organic solvents.

Insulin signaling plays a pivotal role in the regulation of salt chemotaxis learning. DAF-2c encodes an insulin receptor isoform localized in the neuronal axon, regulates the neuronal plasticity of the NaCl-sensing sensory neuron ASER, and a *daf-2c* mutant exhibits substantial defects in salt chemotaxis learning [43] (Fig 8D). Organic solvent exposure promoted the defect in NaCl avoidance or caused defects in attraction in nematodes with a *daf-2c* mutant background, suggesting that organic solvent exposure affects sensory processing, at least in part, independent of DAF-2c signaling in salt chemotaxis learning (Fig 8D).

## Discussion

To reduce the health hazards of industrial chemicals, which are continuously increasing, it is important to fundamentally understand their toxic effects on organisms. Due to its fast lifecycle and proliferative capacity, *C. elegans* are useful for high-throughput screening, including toxic chemical screening [10,11,44]. In this study, a high-throughput method was developed to assess the acute toxicity of organic solvents. The nematodes were exposed to chemicals by soaking in a buffer and were transferred onto an agar plate to capture a 1-min movie. The behaviors of the nematodes were then analyzed using the commercial tracking software, WormLab. This method is simple and does not require special skills. The behavioral toxicity of 30 organic solvents commonly used in industries was comprehensively analyzed using this method. Exposure to any of the organic solvents gradually decreased movement and locomotion was eventually halted. The paralyzed nematodes were recovered by soaking in a buffer without organic solvents, and locomotion impairments were reversible. When the locomotion speed was used as an endpoint, the extent of behavioral toxicities was proportional to the chemical's lipid solubility. This relationship is reminiscent of the Meyer–Overton relationship in humans, in which there is a positive correlation between the anesthetic effects and the lipid solubility of inhalation anesthetics, such as halothane and isoflurane [7]. It has been reported that there is a positive relationship between volatile anesthetics and impaired locomotion in *C. elegans* [15]. Therefore, this study confirmed the feasibility of assessing the general anesthetic effects of industrial chemicals using *C. elegans*.

Exposure to organic solvents, such as short-chain ketones and ethers, caused body shrinkage followed by relaxation, similar to the effects of levamisole, an anthelmintic agent. Levamisole is a well-known potent agonist of acetylcholine receptors, including UNC-29, in *C. elegans* [30]. As previously reported, the *unc-29* mutant is resistant to locomotion impairment caused by exposure to levamisole. Conversely, the *unc-29* mutant was sensitive to organic solvents, including alcohol, ketone, ether, and acetate esters, and required lower concentrations of these organic solvents to induce locomotion impairment than the wild type. The finding that the *unc-29* mutant exhibited body shrinkage after exposure to short-chain ketones or ethers suggests that an unknown molecule(s), other than UNC-29, is a major effector of body shrinkage caused by organic solvents. It has been shown that treatment with ethanol on an agar plate caused body shrinkage via acetylcholine signaling. UNC-63, a subset of the nicotinic acetylcholine receptor, but not UNC-29, is required for ethanol-induced body shrinkage [45]. Further investigation is required to reveal the molecular mechanisms underlying ketone- or ether-induced body shrinkage.

In addition to the general effect of organic solvents on behavior, it was found that organic solvents have specific effects on behavioral patterns, producing different qualitative effects on the amplitude during undulatory locomotion based on the differences in the chemical's functional groups. Alcohols and cellosolves, which have hydroxyl groups, and ketones and acetate

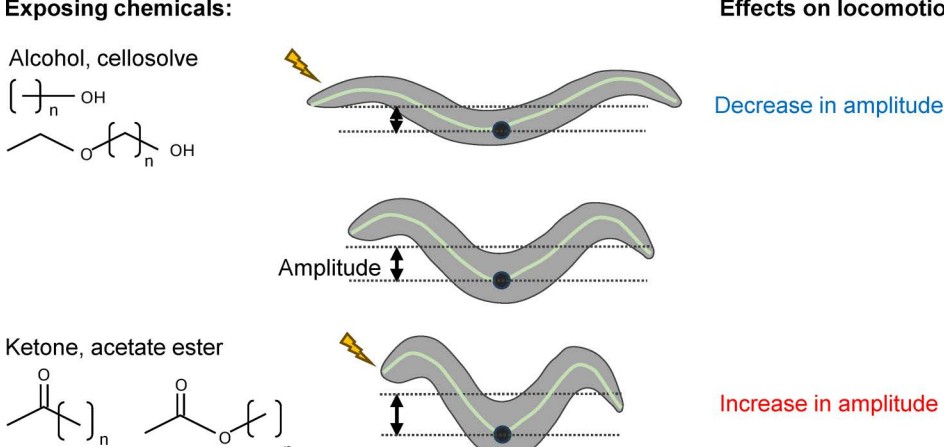

**Fig 9. Summary of the effects of organic solvents on body bending amplitude during locomotion.** Organic solvents affect the amplitudes of body bending during locomotion in opposite ways depending on the chemical's functional groups. Exposure to short-chain alcohols and cellosolve, which have a hydroxyl group, decreases the amplitude of body bending (top), whereas exposure to short-chain ketones and acetate esters, which have a ketone group, increases the amplitude of body bending (bottom).

esters, which have ketone groups, decrease and increase the body bend amplitude during locomotion, respectively (Fig 9). No significant increase in body bending amplitude was observed in the *tax-4* mutant. TAX-4, a cyclic nucleotide-gated channel, plays a pivotal role in sensory transduction in chemosensory neurons, including the AWC olfactory neuron, which senses a variety of volatile chemicals, such as 2-butanone [24]. *In vivo* calcium imaging experiments have shown that a decrease in external odorant concentration activates AWC within seconds [35]. AWC senses constantly changing odor concentrations and produces probabilistic responses, such as changes in turning frequency and turning angle during chemotaxis behavior [35,46]. Meanwhile, more than 5 min of exposure to organic solvents was required to increase the amplitude of body bending, which lasted for at least 1 min. These findings are consistent with the notion that prolonged sensory input from olfactory neurons, such as AWC, induces a long-lasting increase in the amplitude of body bending in a manner different from that in the regulation of chemotaxis. Alternatively, I cannot exclude the possibility that TAX-4-mediated signaling is prerequisite for amplitude alteration after prolonged exposure to organic solvents. Further studies are required to elucidate the molecular and cellular mechanisms of the different qualitative effects of organic solvents on behavioral patterns and their possible conservation across species.

I examined the effects of exposure to NaCl and glucose, which are less toxic, water-soluble chemicals, and compared these with the effects of exposure to organic solvents. Fifteen minutes of exposure to >515 mOsm solutions containing NaCl or glucose caused a marked decrease in locomotion. The locomotion speed reduction was recovered after a subsequent 105-min exposure (total of 120 min of exposure) in the same solution. The $MC_{<50}$ (mM) values of organic solvents with high hydrophilicity, such as methanol, ethanol, and acetone, were higher than those of glucose. These organic solvents are amphiphilic and can penetrate the cell membrane via passive transport. The penetration of ethanol into the *C. elegans* body was experimentally confirmed, and ethanol was shown to act on the neuronal BK potassium channel, followed by inhibition of neuronal activity [20]. In contrast, organic solvents with higher lipid solubility were more toxic, and prolonged exposure to these chemicals did not recover the decreased motility. For example, the $MC_{<50}$ (mM) value of benzyl alcohol is more than 10

times smaller than those of ethanol, NaCl, and glucose. Although the mechanism of action of organic solvents with high lipid solubility is unclear, it is possible that they bind proteins or lipids within the *C. elegans* body and exert adverse effects on motility. Methanol and methyl cellosolve have lower lipid solubility than ethanol, and prolonged exposure to these chemicals recovers slow motility after short-term exposure, as observed after exposure to NaCl or glucose. Therefore, it is possible that exposure to methanol, methyl cellosolve, and other organic solvents causes adverse effects on motility due to dehydration from hyperosmotic pressure to some extent.

Organic solvents cause defects in learned behavior termed "salt chemotaxis learning" at concentrations lower than those that affect locomotion speed. Nematodes respond to water-soluble chemicals, including NaCl, mainly by chemosensory neurons called ASE. ASE neurons extend neuronal processes to the nose tip of the head, and the ciliated ends are exposed to the outside environment through a sensory organ called the amphid [33]. There are molecular mechanisms that regulate sensory transduction and sensory processing in the ciliated ends of ASE [33]. Organic solvents may directly affect the mechanisms for sensory transduction and processing in the ASE cilia, which are directly exposed to the environment. It has been reported that the sensory cilia are targets of 3-octanone, a toxin produced by the basidiomycete oyster mushroom *Pleurotus ostreatus*. Nematodes, including *C. elegans* are paralyzed when in contact with the fungal hyphae of *Pleurotus ostreatus*, which contains 3-octanone [47]. 3-octanone disrupts the cell membrane integrity of multiple *C. elegans* tissues, resulting in cell death caused by extracellular calcium influx into the cytosol and mitochondria. Several mutants lacking cilia are resistant to paralysis and cytotoxicity caused by contact with *Pleurotus ostreatus* or 3-octanone, and cilia are the primary target of 3-octanone [47,48]. In the present study, I demonstrated that low-dose organic solvents cause defects in salt chemotaxis learning, whereas they did not cause gross abnormalities in NaCl sensation. Therefore, organic solvents may have suppressive effects on the mechanisms of neuronal plasticity after NaCl sensation. It is important to understand the cellular and molecular mechanisms underlying the effects of organic solvent exposure in future studies to elucidate the mechanisms of responses to toxic chemicals in learned behavior.

It has been reported that anesthetic exposure affects several behaviors, such as male mating, egg laying, mechanosensation, and chemotaxis, in *C. elegans* [49]. In general, distinct mechanisms are used to exert different types of behaviors. Thus, observation of multiple behaviors provides an opportunity to understand multiple aspects of toxic effects on the nervous system. In addition to innate behaviors, several paradigms of learned behavior have been developed in *C. elegans*. For example, *C. elegans* memorize external environmental cues such as chemicals and temperature during feeding and show learned attractive responses to these cues, which last for several hours [40,50,51]. Nematodes also form long-term memory that lasts for >24 h through repeated training of associations between chemical cues and feeding experience [52]. In future studies, it will be interesting to investigate the effects of toxicants on these complex behaviors and the underlying molecular and cellular mechanisms. These studies can then be applied to the toxicity assessment of higher organisms with complex nervous systems. As a limitation of research using *C. elegans*, the lifespan of wild-type *C. elegans* is 2–3 weeks under standard laboratory conditions, making it difficult to study the effects of chemical exposure over months or years in one generation.

The high-throughput toxicity assay developed in this study can also be used to test the toxicity of environmental pollutants, such as volatile organic compounds (VOCs), heavy metals, and nanoparticles. Toluene, a VOC, has adverse effects on locomotion differently than alcohol, and the underlying molecular mechanism is postulated to be different from that of alcohol in *C. elegans* [53,54]. The toxic effects of heavy metals and metal nanoparticles on the nervous

and reproductive systems have been studied in *C. elegans* [55]. The effects of environmental chemicals, such as bisphenol A and nanoplastics, which are widely present in the environment, on nematodes are also being studied [44,56,57]. Hydrocarbons, such as toluene, are insoluble in water, making it difficult to expose nematodes to these chemicals in a buffer solution as reported in this study. By using methods for exposing toxicants from air [20,54], high-throughput assessments of chemical toxicities can be performed more broadly in the future.

## Supporting information

**S1 Fig. Toxicity of alcohols on *C. elegans* locomotion.** Immobility (A) and locomotion speed (B) after exposure to alcohols for 15, 30, 60, or 120 min. Each data point represents the mean ± the standard error of the mean (SEM). Locomotion speed was normalized to the average value for untreated control.
(TIF)

**S2 Fig. Toxicity of ketones, acetate esters, and ethers on *C. elegans* locomotion.** Locomotion speed after exposure to ketones (A), acetate esters (B), or ethers (C) for 15, 30, 60, or 120 min. Each data point represents the mean ± the standard error of the mean (SEM). Locomotion speed was normalized to the average value for untreated control.
(TIF)

**S3 Fig. Toxicity of cellosolves and cellosolve acetate on *C. elegans* locomotion.** Locomotion speed after exposure to cellosolves or a cellosolve acetate for 15, 30, 60, or 120 min. Each data point represents the mean ± the standard error of the mean (SEM). Locomotion speed was normalized to the average value for untreated control.
(TIF)

**S4 Fig. Correlation between the toxicity of ketones or cellosolves and their carbon number.** Relationship between the behavioral toxicity ($MC_{<50}$) after 1 h of exposure to ketones (A) or cellosolves (B) and the carbon number of the chemicals. Locomotion speed is used as an endpoint. $R^2$ values were determined based on simple linear regression analysis.
(TIF)

**S5 Fig. Correlation between the toxicity of organic solvents and their lipid solubility.** Relationship between the $MC_{<50}$ after exposure to organic solvents for 15, 30, or 120 min and the octanol–water partition coefficient (log Kow), which reflects the lipid solubility of organic solvents. Locomotion speed is used as an endpoint. $R^2$ values were determined based on simple linear regression analysis.
(TIF)

**S6 Fig. Relationship between the toxicity of organic solvents and their melting point.** Relationship between the $MC_{<50}$ after exposure to organic solvents for 1 h and the organic solvent's melting point. Locomotion speed is used as an endpoint. *P* and $R^2$ values are determined based on simple linear regression analysis, and no significant correlation is observed ($P = 0.1913$).
(TIF)

**S7 Fig. Effects of levamisole on the locomotion and body length of *C. elegans*.** (A–C) Immobility (A), locomotion speed (B), and body length (C) after exposure to levamisole for 15, 30, 60, or 120 min. Each data point represents the mean ± the standard error of the mean (SEM). (D) Representative images of nematodes on agar plates after exposure to different concentrations of levamisole for 120 min.
(TIF)

**S8 Fig. 2-butanone and tetrahydrofuran cause body shrinkage followed by relaxation.** Representative images of nematodes on agar plates after exposure to different concentrations of 2-butanone (A) or tetrahydrofuran (B) for 120 min.
(TIF)

**S9 Fig. Changes in body length after exposure to organic solvents.** The mean values ± the standard error of the mean (SEM) of the body lengths of nematodes after 1 h exposure to organic solvents (red) or without exposure (blue). The exposure concentrations are the $MC_{<50}$ values, which were determined by the locomotion speed after 1 h of exposure. Data were normalized to the average values of the no-exposure control. $*P < 0.05$, unpaired *t*-test with Holm–Sidak correction.
(TIF)

**S10 Fig. Chemical toxicity on locomotion in an *unc-29* mutant.** Immobility (left column) and locomotion speed (right column) of *unc-29(e193)* after exposure to levamisole (A), 2-butanone (B), or tetrahydrofuran (C) for 15, 30, 60, or 120 min. Each data point represents the mean ± the standard error of the mean (SEM). Locomotion speed was normalized to the average value for untreated control (B, C).
(TIF)

**S11 Fig. The *unc-29* mutant is sensitive to organic solvent toxicity.** (A, B) Relationship between the $MC_{<50}$ after organic solvent exposure for 1 h and the octanol–water partition coefficient (log Kow), which reflects the lipid solubility of an organic solvent. Immobility (A) and locomotion speed (B) are used as the endpoints in the wild type (blue) and *unc-29(e193)* (red). (C) Violin plots of the body lengths of *unc-29(e193)* mutants after exposure to organic solvents. The exposure concentrations are same as those used for the analysis in the wild type (Fig 3B). Data were normalized to the average values of the no-exposure control. Each dot represents the body length of a nematode after exposure (red) or without exposure (blue). $*P < 0.05$, unpaired t-test with Holm–Sidak correction.
(TIF)

**S12 Fig. Time course of recovery from paralysis after organic solvent exposure.** Locomotion speed after recovery in a buffer solution after exposure to an organic solvent for 1 h (A), 30 min (B), or 15 min (C) are shown. The exposure concentrations of the organic solvents are the minimum concentrations that cause complete paralysis. Each data point represents the mean ± the standard error of the mean (SEM).
(TIF)

**S13 Fig. Changes in the amplitudes during locomotion after organic solvent exposure.** Violin plots of the amplitudes after soaking in a buffer containing organic solvents for 15 min. Each dot represents the mean amplitude during each track. $*P < 0.05$, one-way ANOVA with Dunnett test, compared with the no-exposure control.
(TIF)

**S14 Fig. Locomotion speed after exposure to organic solvents in *tax-4* and *osm-9* mutants.** Locomotion speed of *tax-4(p678)* (A, red bars), *osm-9(ky10)* (B, red bars), and the wild-type (blue bars) nematodes after exposure to organic solvents for 1 h. Each bar represents the mean ± the standard error of the mean. $*P < 0.05$, one-way ANOVA with Dunnett test, compared with the no-exposure control.
(TIF)

**S15 Fig. *osm-9* mutants show changes in amplitude of the body bends after exposure to organic solvents comparable to the wild type.** Violin plots of the amplitudes after soaking in

a buffer containing organic solvents for 15 min in *osm-9(ky10)* nematodes. Each dot represents the mean amplitude during each track. *$P < 0.05$, one-way ANOVA with Dunnett test or Welch's t-test (for Ethyl acetate), compared with the no-exposure control.
(TIF)

**S16 Fig. Effects of NaCl exposure on motility and body length.** Locomotion speed (left) and body length (right) of the wild type (A) and the *dyf-11(pe554)* mutant (B) after exposure to sodium chloride for 15, 30, 60, or 120 min. Each bar or data point represents the mean ± the standard error of the mean. *$P < 0.05$, one-way ANOVA with Dunnett test, compared with the no-exposure control.
(TIF)

**S17 Fig. Comparison between effects of chemical exposure for 15 and 120 min on locomotion speed.** Locomotion speed after exposure to indicated chemicals for 15 or 120 min. Each bar represents the mean ± the standard error of the mean. *$P < 0.05$, unpaired t-test with Holm–Sidak correction. Locomotion speed was normalized to the average value for untreated nematodes.
(TIF)

**S18 Fig. Salt chemotaxis learning assay.** (A) Procedure for the salt chemotaxis learning assay. (B) Format for the salt chemotaxis test and calculation of the chemotaxis index. (C) Size of a chemotaxis test plate.
(TIF)

**S19 Fig. Determination of *C. elegans* body length and width.** (A) The body length of a nematode was determined as the length from head to tail along the central axis (blue line). The body width was determined as the average length of cross-sections (red lines) over the entire body. (B) The average body length (left, 1049 μm) and the average body width (right, 92.30 μm) were determined based on body lengths and widths of 650 nematodes, calculated using Wormlab.
(TIF)

**S1 File. Raw data of the main figures.**
(XLSX)

**S2 File. Raw data of the supporting figures.**
(XLSX)

**S3 File. Raw statistics of the main figures.**
(XLSX)

**S4 File. Raw statistics of the supporting figures.**
(XLSX)

**S1 Movie. Representative movie of tracking the nematodes moving on agar plates using the WormLab software.** Nematodes were exposed to a buffer without organic solvent for 1 h before recording the movement.
(MP4)

**S2 Movie. Representative movie of tracking the nematodes moving on agar plates using the WormLab software.** Nematodes were exposed to buffer, including 0.5% benzyl alcohol, for 1 h before recording the movement.
(MP4)

**S3 Movie. Representative movie of the locomotion of nematodes with decreased body bending amplitudes after exposure to 3% isopropanol for 15 min.**
(MP4)

**S4 Movie. Representative movie of the locomotion of nematodes with increased body bending amplitudes after exposure to 2% 2-butanone for 15 min.**
(MP4)

**S5 Movie. Representative movie of the locomotion of nematodes with increased body bending amplitudes after exposure to 2% tetrahydrofuran for 15 min.**
(MP4)

## Acknowledgments

I would like to thank Drs. Tatsushi Toyooka, Rui-Sheng Wang, Yukie Yanagiba, and Makiko Nakano at the National Institute of Occupational Safety and Health, Japan, for their helpful discussions and support. The *C. elegans* and *E. coli* strains were provided by Dr. Yuichi Iino's lab and the CGC, which is funded by NIH Office of Research Infrastructure Programs (P40 OD010440).

## Author contributions

**Conceptualization:** Masahiro Tomioka.

**Data curation:** Masahiro Tomioka.

**Formal analysis:** Masahiro Tomioka.

**Investigation:** Masahiro Tomioka.

**Methodology:** Masahiro Tomioka.

**Project administration:** Masahiro Tomioka.

**Resources:** Masahiro Tomioka.

**Validation:** Masahiro Tomioka.

**Writing – original draft:** Masahiro Tomioka.

**Writing – review & editing:** Masahiro Tomioka.

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
