## [Decision Letter · Decision Letter 0]

11 Nov 2024

PONE-D-24-41559

Observation of general and specific adverse effects of organic solvents on *Caenorhabditis elegans* by toxicity testing using behavioral analysis

PLOS ONE

Dear Dr.  Tomioka,

Thank you for submitting your manuscript to PLOS ONE. After careful consideration, we feel that it has merit but does not fully meet PLOS ONE’s publication criteria as it currently stands. Therefore, we invite you to submit a revised version of the manuscript that addresses the points raised during the review process.

We look forward to receiving your revised manuscript.

Kind regards,

Myon-Hee Lee, Ph.D

Academic Editor

PLOS ONE

Journal Requirements:

Reviewers' comments:

Reviewer's Responses to Questions

**Comments to the Author**

1. Is the manuscript technically sound, and do the data support the conclusions?

Reviewer #1: No

Reviewer #2: Yes

2. Has the statistical analysis been performed appropriately and rigorously?

Reviewer #1: Yes

Reviewer #2: Yes

3. Have the authors made all data underlying the findings in their manuscript fully available?

Reviewer #1: Yes

Reviewer #2: Yes

4. Is the manuscript presented in an intelligible fashion and written in standard English?

Reviewer #1: Yes

Reviewer #2: Yes

5. Review Comments to the Author

Reviewer #1: Tomioka presents a nice study on how 30 different organic solvents that are toxic to humans and other animals change behavioral endpoints in C. elegans. They determine dose response curves for locomotory parameters using Wormlab as well as an associative learning paradigm and chemotaxis. Strengths include abundant data, simple and clearly graphed figures. Weaknesses include assumptions made that these behaviors reflect direct adverse action of these toxins on neurons and/or muscles, lack of context in citations, lack of evidence that these toxins penetrated the worms and at what concentration. I look forward to seeing this published.

Major issues:

1. The author repeatedly uses the phrase “adverse effects” and clarify their assumption that responses to the solvents represent a passive response of the solvents on the neurons and/or muscles. However, they do not consider to what extent these responses may reflect a mixture of that as well as the following: a) active attempts by the worm to escape the toxin by increasing speed via body curvature, b) active attempts to slow down due to attraction to the solvent (see PMID: 8348618), c) passive yet reservable responses by the worm to dehydration by the solvent (see PMID: 15166144). Rather than “adverse”, I’d recommend a neutral term such as behavioral response or change. Even for salt attraction plasticity, it seems adaptive rather than maladaptive to not form an association when danger is apparent. So, this isn’t obviously an adverse effect in my opinion.

2. How much of chemicals penetrate the worm given the amount, duration of treatment, and buffer? I don’t expect the author to make measured estimates for internal concentrations for each of the 30 solvents and at each dose. But it would be appropriate for them to measure some and to cite other papers that have measured these parameters already for some chemicals (e.g. gas chromatography or colorimetric assays) (see PMID: 22486589). For the goal of their work to be relevant to human toxicity, it is important to know if internal concentrations that elicit behavioral endpoints in worm relate to toxicity in human tissues.

3. The author says that the site of action of toxicants have not been extensively studied (line 77), but it has for many chemicals. The Morgan lab has published many papers on how solvents act or do not act in expectation with the Meyer-Overton rule (please look for many including this one from 1995, “Overall, we think that drawbacks of C. elegans are out-weighed by the animal’s impeccable adherence to the Meyer-Overton relationship, the reversibility of our anesthetic endpoint, the qualitative behavior of the animals in various anesthetic agents, and the maintenance of a cut-off effect." PMID: 8749805, also 1990 PNAS paper PMID: 2326259). The Morgan lab identified a broad array of genes that are required to respond to specific solvents, breaking the M-O rule. Likewise, many papers have been published on protein and lipid targets of ethanol (and toluene) with quantitative analysis of worm posture and motion that are not cited and put into context here (e.g. many, but see PMID: 14675531, PMID: 15182714, PMID: 21945072, PMID: 22574115, PMID: 25342716). A suggestive amino acid target for ethanol was shown in PMID: 25031399. The author has conducted a wide range of experiments. Please put your hard work into context.

4. Repeating an important idea above, just because the worm shrinks and recovers its body size with levamisole exposure, doesn’t mean that if reversable body shrinkage is also observed with a solvent that it happens the same way. The author shows no experimental evidence that the underlying mechanisms are the same. However, there are ways to do this by testing mutants deficient in an array of muscle receptors or ion channels. I worry to what extent this shrinking reflects the worm becoming dehydrated. Would osr-1 or an aquaporin mutant resist this shrinkage? Also, please discuss your UNC-29 results and body shrinkage in context with PMID: 25342716.

Minor issues:

1. The final supplementary cartoon is speculative. They have no evidence that sensory neurons are affected at low concentrations (the worms chemotax fine). We do not know if muscles are less affected than motorneurons. I recommend removing this figure.

2. Use first person to help clarify what you did versus others. This is especially important when you mention other people’s work and then your own - both using third person in subsequent sentences. E.g. lines 116, 290

3. Line 53. The chemicals might have effects not just by penetrating cells, but also on cells or on excreted components.

4. Given that worms respond to ethanol very differently with different buffers (see PMID: 22486589), please make sure to mention which buffer was used. Also clarify if worm was exposed to chemical in liquid buffer or with chemical in the agar substrate with air above, which also causes different drug effects. Note supply source for agar.

5. Please make it more apparent how the immobile state is defined. How is this different than the pause state?

6. Figure 1: please add units to y-axis, adjust fonts for all graphs in all panels of all figures to be more uniform and readable. For B, the situation is hard to understand without additional information. "Soaking in a buffer" on the x axis suggests that worms were soaked thrashing in buffer BEFORE measuring their crawling speed on dry agar plates. Is this the case? Explain when worms were soaked in buffer, for how long, and when did recording behavior happen. What was the buffer? Were there more than one buffer? Why not make the pause area to be a gray box rather than “colorless”? In legend, why is buffers plural here? Was there more than one type of buffer? If so, explain. Lastly, "exposure without bz" is confusing as written. It is more accurate and simpler to write "average value for untreated worms"

7. Figure 2: rate? per unit time? Isn't this a proportion? Figure 2 is confusingly laid out. Please label all panels A-D. Confusing on what end point(s) is for B and C. The author has an opportunity to test for significant change in R^2 value for these models in B and C. Might be able to say that lipid solubility is significantly better model.

8. Line 331: recovering in liquid? Or on agar plate? Please clarify. What buffer?

9. Line 357: Please cite the original data in PMID: 14675531 and PMID: 15182714

10. Line 365: please put into context with how worm can respond to these chemicals with the use of smell. See 1993 Cell paper PMID: 8348618

11. Line 376: it is hard to conclude this here. The effect could also be on muscle tone, on hydration, and/or even on their behavioral excitement or repulsion responses that one would expect to change their posture.

12. Line 414; clarify here that you know that the solvent treatment did not impact sensation or locomotion because your control group could still move to salt.

13. I do not understand why the author tested eat-4 as a control. They cannot perform plasticity. What was to be learned?

14. Line 490: keep in mind that even if the chemicals cannot penetrate the worm, it can smell these chemicals and change its locomotory behavior in response. This would be reflected in changes in body posture.

15. Line 528-530: this doesn't appear to be the general case because the solvents did not affect normal chemotaxis. The solvents somehow interrupted the plasticity mechanisms. Actually it is remarkable that they could chemotaxis fine after exposure to these chemicals, because like you write, the ASE neurons should be affected – presumably negatively. But they chemotaxed fine. Perhaps it is because worm has adapted to live in compost with rotting material and organic solvents?

16. Line 542. a number of relevant studies on toxicity are not cited and put in context. Use pubmed to search for "elegans toxic organic solvent" or more

Reviewer #2: In the current study, the authors used C. elegans as animal model to perform a high-throughput assessment of general and specific adverse effects of organic solvents on behaviors, including both locomotion and chemotaxis-based learning. The specific comments for this MS are as follows:

1. Title: Suggest change it to: “High-throughput assessment of general and specific adverse effects of organic solvents on behaviors in Caenorhabditis elegans”.

2. Abstract: Please specify the chemical showing toxic effects on behavior. In addition, the examined doses of chemicals should be indicated.

3. Discussion: In this study, besides locomotion, the learning was also assessed. Please discuss the possible value and limitation of developed method while assessing other types of complex behaviors.

4. Discussion: During the past 10 years, besides for molecular toxicology, one of important value of C. elegans is used to assess pollutants at environmentally relevant concentrations (ERCs). Please discuss the possible value of developed method at this aspect.

6. PLOS authors have the option to publish the peer review history of their article (what does this mean? ). If published, this will include your full peer review and any attached files.

**Do you want your identity to be public for this peer review?** For information about this choice, including consent withdrawal, please see our Privacy Policy .

Reviewer #1: No

Reviewer #2: No

---

## [Author Response · Author response to Decision Letter 1]

20 Feb 2025

Thank you for reviewing my manuscript and providing me with constructive and insightful comments on the manuscript. I performed experiments to address the reviewers’ concerns and thoroughly revised the manuscript according to the reviewers’ comments. Furthermore, the manuscript has been edited for English language, grammar, punctuation, and spelling by a proofreading service, Enago. My point-by-point response to the comments is uploaded as a file labeled "Response to Reviewers".

---

## [Decision Letter · Decision Letter 1]

10 Mar 2025

High-throughput assessment of the behavioral responses to toxic organic solvents in Caenorhabditis elegans

PONE-D-24-41559R1

Dear Dr. Tomioka

We’re pleased to inform you that your manuscript has been judged scientifically suitable for publication and will be formally accepted for publication once it meets all outstanding technical requirements.

Kind regards,

Myon-Hee Lee, Ph.D

Academic Editor

PLOS ONE

Additional Editor Comments (optional):

Reviewers' comments:

Reviewer's Responses to Questions

**Comments to the Author**

1. If the authors have adequately addressed your comments raised in a previous round of review and you feel that this manuscript is now acceptable for publication, you may indicate that here to bypass the “Comments to the Author” section, enter your conflict of interest statement in the “Confidential to Editor” section, and submit your "Accept" recommendation.

Reviewer #2: (No Response)

2. Is the manuscript technically sound, and do the data support the conclusions?

Reviewer #2: (No Response)

3. Has the statistical analysis been performed appropriately and rigorously?

Reviewer #2: (No Response)

4. Have the authors made all data underlying the findings in their manuscript fully available?

Reviewer #2: (No Response)

5. Is the manuscript presented in an intelligible fashion and written in standard English?

Reviewer #2: (No Response)

6. Review Comments to the Author

Reviewer #2: The authors have well explained the raised questions raised by me. Thus, my final comment for this revised MS is: Accept.

7. PLOS authors have the option to publish the peer review history of their article (what does this mean? ). If published, this will include your full peer review and any attached files.

**Do you want your identity to be public for this peer review?** For information about this choice, including consent withdrawal, please see our Privacy Policy .

Reviewer #2: No

---

## [Editor Report · Acceptance letter]

PONE-D-24-41559R1

PLOS ONE

Dear Dr. Tomioka,

I'm pleased to inform you that your manuscript has been deemed suitable for publication in PLOS ONE. Congratulations! Your manuscript is now being handed over to our production team.

Kind regards,

on behalf of

Dr. Myon-Hee Lee

Academic Editor

PLOS ONE